# Controlled release testing of the static chamber methodology for direct measurements of methane emissions

James P. Williams[1], Khalil El Hachem[1], and Mary Kang[1]

[1]Department of Civil Engineering, McGill University, Montréal, Québec, Canada

**Correspondence:** James P. Williams (philip.williams@mail.mcgill.ca)

**Abstract.** Direct measurements of methane emissions at the component level provide the level of detail necessary for developing actionable mitigation strategies. As such, there is a need to understand the magnitude of component level methane emission sources and test methane quantification methods that can capture methane emissions at the component level used in national inventories. The static chamber method is a direct measurement technique that is being applied to measure large and complex methane sources such as oil and gas infrastructure. In this work we compile methane emission factors from the IPCC emission factor database to understand the magnitude of component level methane flowrates, review the tested flowrates and measurement techniques from 40 controlled release experiments, and perform 64 controlled release testing of the static chambers methodology with mass flowrates of 1.02, 10.2, 102, and 512 grams of methane per hour (g/hour). We vary the leak properties, chamber shape, chamber size, and usage of fans to evaluate how these factors affect the accuracy of the static chamber method. We find that 99% of component level methane emission rates from the IPCC emission factor database are below 100 g/hour, and that 77% of previously-available controlled release experiments did not test for methane mass flowrates below 100 g/hour. We find that the static chamber method quantified methane flowrates with an overall accuracy of +14/-14%, and that optimal chamber configurations (i.e., chamber shape, volume, usage of fans) can improve accuracy to below $\pm 5\%$. We find that smaller chambers ($\leq$20 L) performed better than larger volume chambers ($\geq$20 L), regardless of the shape of chamber or usage of fans. However, we found that the usage of fans can substantially increase the accuracy of larger chambers, especially at higher mass flowrates of methane ($\geq$100 g/hour). Overall, our findings can be used to engineer static chamber systems for future direct measurement campaigns targeting a wide range of sources, including landfills, manholes, and oil and natural gas infrastructure.

## 1 Introduction

Methane is a potent greenhouse gas, and international initiatives such as the Global Methane Pledge (Global Methane Pledge, 2021.) have motivated national commitments towards reducing emissions of methane from a variety of sectors from waste to energy to agriculture. In order to materialize methane reductions through actionable mitigation strategies, accurate methane inventories that quantify methane from different sectors and sources are needed. Methane emission sources can be broadly classified as either site or component level emissions, where site level emissions are the sum of multiple emitting components. There are also additional classifications such as facility, regional, continental, and global level (NACEM., 2018) which

encompass each preceding classification within a larger agglomeration of methane emission sources. Understanding methane emissions at the component level (i.e., the smallest tier of methane emissions sources) is particularly important for developing actionable methane reduction strategies as these data can be used to directly analyze the cost-benefits of mitigation options which allows policy makers and project developers to make informed decisions (Kang et al., 2019; IEA, 2021). Therefore, it is important that we test and develop methane quantification methods that are capable of measuring methane emissions accurately at the component level.

To select the optimal methane measurement methods, there is a need to understand the expected magnitude of methane emission rates from different component level sources. Some data sources such as the IPCC emission factor database (IPCC-EFDB, 2022) have compiled emission factors for different greenhouse gas sources around the world. However, some emission factors within this database are provided at the site level, and some are provided in alternative forms to methane emission rates (e.g., mass of methane emitted per ton of waste) which makes it difficult to determine the magnitude of expected component level emission rates. As such, our goal is to determine the approximate magnitude of methane emission rates at the component level so that we can conduct tests at appropriate methane flowrates.

There are multiple methods that are used to quantify methane emissions, which we classify here as either indirect or direct methods. Indirect methane quantification methods are based on measurements made away from the source of emissions and can often be conducted without site access. These methods include mobile surveying, stationary tower (e.g., eddy covariance tower) measurements, aerial based surveys, and satellite measurements (Cusworth et al., 2022; Edie et al., 2020; Robertson et al., 2017; Kumar et al., 2022; Riddick et al., 2022; Ravikumar et al., 2017; Ayasse et al., 2019; Cooper et al., 2021; Varon et al., 2018; de Foy et al., 2023). Direct methane quantification methods are based on quantifying methane emissions directly at the source of emissions and generally require site access. The most common direct measurement methods include optical gas imaging cameras, Hi-Flow samplers, and chamber based methodologies.

Methane sources can be classified as component, site, facility, regional, and global level sources in order of increasing spatial scales (NACEM., 2018). As an example, a valve on an oil and gas well would constitute a component level source whereas all oil and gas wells in the Appalachian basin would comprise a regional methane source. The advantages of methane inventories created from component level measurements are high resolution and easy comparisons to regional inventories, which are predominantly made using component level data (EPA, 2021; ECCC, 2021), where specific discrepancies can be identified (Rutherford et al., 2021). Indirect measurements can be used to measure methane emissions at site/facility/regional levels. On the other hand, direct measurement methods are labour intensive and can omit methane sources when scaling up measurements to facility/regional/global levels, but can quantify and attribute methane emissions at the component level. In terms of testing methane measurement methods for accuracy, the majority of published literature has focused on indirect methods (e.g., Robertson et al. (2017); Edie et al. (2020); Sherwin et al. (2021); Aubrey et al. (2013)), whereas few studies have tested and quantified the accuracy of direct measurement methods (Riddick et al., 2022; Pihlatie et al., 2013; Christiansen et al., 2011).

Among the direct measurement methods, optical gas imaging cameras and Hi-Flow samplers both have limits of detection at roughly 20 g/hour (Ravikumar et al., 2017; Fox et al., 2019). However the stated uncertainties of optical gas imaging cameras

in Fox et al. (2019) of 3-15% are noted as being complex and likely much higher, and there have been several studies that have highlighted measurement errors attributed to the Hi-Flow sampler (Connolly et al., 2019; Howard et al., 2015). As an alternative, the static chamber methodology is a well-established direct methane measurement method (Riddick et al., 2022; Pihlatie et al., 2013) traditionally used in the measurement of methane and other trace gas emissions emissions from soils (Conen et al., 1998; Raich et al., 1990; Smith et al., 2003). In recent years, the static chamber method has been applied in a wide range of settings such as the quantification of methane emissions from oil and gas wells (Lebel et al., 2020; Williams et al., 2020; Kang et al., 2014; El Hachem et al., 2022; Townsend-Small et al., 2016, 2021; Saint-Vincent et al., 2020; Riddick et al., 2019), manholes (Fries et al., 2018; Williams et al., 2022), landfill vents and observation wells (Williams et al., 2022), and natural gas (NG) distribution infrastructure (Williams et al., 2022; Lamb et al., 2016, 2015). All of these sources vary in terms of their leakage properties and structural complexity with regards to the installation of chambers over leaking components. However, there are few studies that have quantified the measurement accuracy of the static chamber method, and even fewer (Riddick et al., 2022; Lebel et al., 2020) that have tested the static chamber method in conditions that mimic the wide range of settings in which they are now being used.

Different methane sources can emit methane at the same mass flowrates albeit at different volumetric flowrates depending on the methane concentration of the source. For example, biogas produced from landfills ($\sim$50% methane) will differ in its source methane concentration from NG from a distribution pipeline ($\sim$90% methane). To our knowledge, there have been no studies that have tested the effects of a varying volumetric flowrate of methane as a factor to be considered in measurement accuracy for any methane measurement method. In terms of the structural complexity of these sites, several studies have employed large chambers with sub-optimal shapes to accommodate more complex sites. For example, a study by Lebel et al. 2020 in California targeting oil and gas wells used three static chambers that ranged in size (i.e., 33.8 litres to 32,659 litres) and shape (i.e., cylindrical and rectangular configurations). A key assumption in the static chamber method is that the air/gas within the chamber is well mixed (Kang et al., 2014). If the emission rate is low and the chamber is large, it may be challenging to have the gases in the chamber be well-mixed. Chamber shapes such as rectangular have been shown to have "dead zones" where gases are not well-mixed, thereby lowering the effective volume of the chamber (Christiansen et al., 2011).

In this work we: 1) compile component level methane emission factors and categorize them by source category; 2) investigate prior controlled release testing of direct and indirect methane measurement methods to identify gaps in testing; 3) test the impacts of physical factors such as the chamber shape, size, and usage of fans on the accuracy of methane flowrate estimates; and 4) test the effects of leak properties (i.e., mass flowrates, volumetric flowrates, concentration of methane in the leak) on the accuracy of chamber measurements. Our results highlight the applicability of the static chamber technique in direct measurements of methane emissions and provide the detail necessary to inform future measurement campaigns.

## 2  Methodology

We compiled a dataset of methane emission factors from the IPCC emission factor database (IPCC-EFDB, 2022) and categorized them into three source categories: agriculture, forestry, and other land use (AFOLU), energy, and waste. We removed all

**Table 1.** Physical descriptions of chambers used for the controlled release experiments. Qualitative descriptions of chamber volume are indicated in parenthesis in the first row.

| Chamber ID | A | B | C | D |
|---|---|---|---|---|
| Chamber size (L) | 2,265 (large) | 322 (large) | 18 (small) | 14 (small) |
| Shape | Rectangular | Cylindrical | Cylindrical | Rectangular |
| Structure | Collapsible | Collapsible | Solid | Solid |
| Material | PE tarp | PE plastic | HDPE plastic | HDPE plastic |
| Aspect ratio (Height:width) | 5:4 | 18:11 | 1:1 | 4:5 |

PE = polyethylene, HDPE = high density polyethylene

emission factors that were not related to a direct mass flowrates of methane at the component level. We removed all emission factors presented as methane flux rates (i.e., mass of methane emitted over a given area), and where possible, converted all remaining methane emission factors to component level methane mass flowrate presented in grams of methane emitted per hour based on assumptions outlined in the SI - Table S1.

We performed a literature review of 40 controlled release experiments of methane using both indirect and direct methods to evaluate the range of methane flowrates tested and the methods used. The criteria for the literature review included all studies where methane was released at known mass flowrates of methane from above-ground points and excludes studies related to methane released in the subsurface, laboratory experiments of methane plume transport through porous media, and studies where the tested mass flowrates of methane were not reported. We also exclude studies where methane quantification methods were tested on in-situ methane sources for validation. We categorized the studies based on the tested measurement platform which we grouped into eight categories: satellite (indirect method), manned aerial vehicle (indirect method), unmanned aerial vehicle (indirect method), stationary tower (indirect method), mobile surveying (indirect method), Hi-Flow sampler (direct method), camera-based (direct method), chamber measurements (direct method), and/or a combination of all the above.

We performed controlled releases of methane for the static chamber method outdoors on the McGill University campus in Montréal, Canada on June 2nd, 8th, and 10th, 2021. The weather for these days was sunny with sparse clouds with an average temperature of $25^o$C and wind speeds ranging from 5-15 kph (World Meteorological Station ID: 71612) . We designed the controlled release experiments to test a combination of six different factors: mass flowrate, volumetric flowrate, methane percentage of leaking gas, chamber shape (i.e., rectangular versus circular), chamber size (i.e., 14 L, 18 L, 322 L, and 2,265 L), and the usage of the fans within the chamber. For the 322 L and 2,265 L chambers we used four battery-powered equipment cooling fans (airflow: 40 ft$^3$ of air per minute) installed at the top of the chamber framework and oriented at $45^o$ angles downward into the chamber, and for the smaller chambers we used one fan. The tested chamber shapes were a 2,265 L rectangular chamber, a 322 L cylindrical chamber, a 18 L cylindrical chamber, and a 14 L rectangular chamber (Table 1). In addition, for a qualitative comparison between chamber sizes, we define $\leq$20 L chambers as small, and the 322 L and 2,265 L chambers as large. Other factors such as the aspect ratio of the chamber, the rigidity of the chamber material, and the type of chamber material are provided in Table 1.

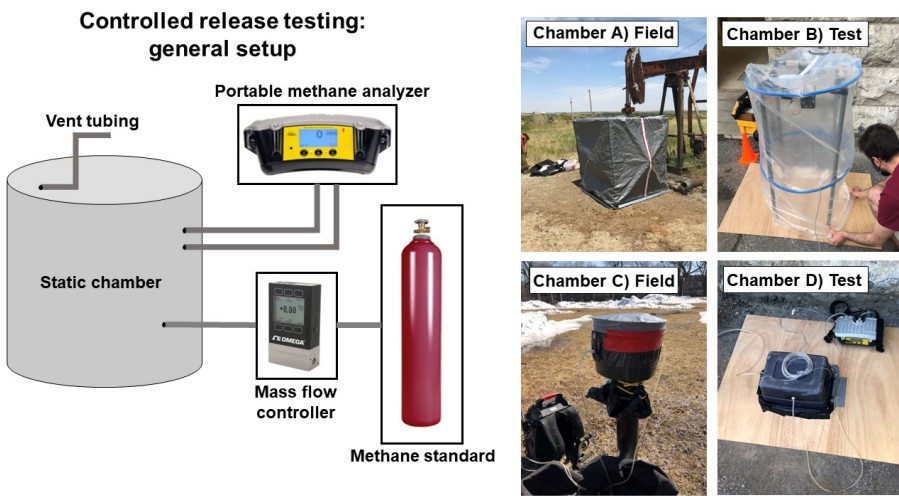

**Figure 1.** Diagram of the controlled release experiments for the static chamber method (left), with photos of chamber deployments in both field settings and during controlled release testing. The four chambers shown correspond to the four chambers we tested.

We tested four different mass flowrates: 1.02 g/hour, 10.2 g/hour, 102 g/hour, and 512 g/hour. In order to provide a qualitative comparison between mass flowrates, we define the mass flowrates of 1.02 and 10.2 g/hour as small flowrates, and the 102 and 512 g/hour releases as high flowrates. At least two different volumetric flowrates and two different methane concentrations were used for each of the mass flowrates we tested. The volumetric flowrates ranged from 0.238 SLPM (standard litres per minute) to 23.8 SLPM for a total of ten unique leaks (Table 2). We controlled mass flowrates of methane using two mass flow controllers (Masterflex Mass Flowmeter Controller) with volumetric flow ranges of 50 to 0.5 SLPM and 1 to 0.01 SLPM (error of $\pm0.8\%$ of reading and $\pm0.2\%$ of full-scale range). Both mass flow controllers were factory calibrated prior to use for these experiments. Four different methane standards, prepared by Linde Canada, were used in our study: 100%, 50%, 10%, and 5% methane ($\pm0.5\%$) all with a gas balance of air.

We performed the controlled release tests by releasing methane through Tygon tubing connected to the chamber. We oriented the tubing to the center of the chamber and secured it to the ground with tape to orient the flow upwards. We measured methane concentrations within the chamber continuously using a Sensit Portable Methane Detector which has a range of 0-100% methane, precision of 1 ppm, sampling frequency of 1 Hz, pump flow of 1 L per minute, and a reported accuracy of $\pm10\%$. The analyzer was located outside the chambers with the analyzer inlets and outlets connected to the chamber ports in a closed loop with Tygon tubing of equal lengths for the inlet and outlet. Chambers were equipped with a 2 meter coil of 1/8" diameter Tygon tubing to allow for pressure equalisation between the chamber and the atmosphere (Christiansen et al., 2011). The duration of each controlled release was 5 minutes, with the exception of releases where fans were used within the chamber and methane concentrations were expected to reach the lower explosive limit of methane (i.e., 5% methane) before the 5 minute mark. Since the fans were not intrinsically safe, these experiments were terminated when the methane concentration within the chamber reached 35,000 ppm (i.e., 70% LEL). For this same reason, we did not test mass flowrates of 102 and 512 g/hour

**Table 2.** Leak properties of ten different leaks used in controlled release experiments including percentage errors associated with mass flow controllers (MFC). Qualitative descriptions of the leak sizes are shown in parenthesis in the first column.

| Methane mass flowrate (g/hour) | Volumetric flowrate (SLPM) | % methane | MFC error |
|---|---|---|---|
| 1.02 (small flowrate) | 0.238 | 10% | ±1.64% |
| - | 0.476 | 5% | ±1.22% |
| 10.2 (small flowrate) | 0.476 | 50% | ±1.64% |
| - | 2.38 | 10% | ±5.00% |
| - | 4.76 | 5% | ±2.90% |
| 102 (large flowrate) | 2.38 | 100% | ±5.00% |
| - | 4.76 | 50% | ±2.90% |
| - | 23.8 | 10% | ±1.22% |
| 512 (large flowrate) | 11.9 | 100% | ±1.64% |
| - | 23.8 | 50% | ±1.22% |

SLPM = Standard litres per minute, MFC = Mass flow controller

with the smaller chambers (i.e., $\leq$20 L) with fans present. When larger volume chambers were used, we were able to maintain methane levels within the chamber at safe limits.

Mass flowrates were calculated from the rate of methane build-up within the chamber over time multiplied by the volume of the chamber (1): where $M$ is the mass flowrate of methane, $dc/dt$ is the change in methane concentration over time, and $V$ is the volume of the chamber.

$$M = \frac{dc}{dt}V \qquad (1)$$

For some experiments, methane concentrations within the chamber were expected to rapidly reach steady-state. Steady-state is reached when methane concentrations no longer increase over time in the chamber and the concentration of methane within the chamber is equal to the concentration of the released gas. The residence time, or time to reach steady-state, is defined by (2): where $\tau$ is the residence time, $V$ is the volume of the chamber, and $Q$ is the volumetric flowrate of gas (i.e., methane and balance gas combined) into the chamber. For any controlled releases where the expected residence time was two minutes or shorter, we only used the initial ten data points for the linear regression to avoid the period of exponential decay as methane concentrations approach steady-state (Pihlatie et al., 2013).

$$\tau = \frac{V}{Q} \qquad (2)$$

We summarized the results of the controlled release tests by calculating the percentage deviation of the the true versus measured methane flowrate (3): where $E$ is the error in (%), $Q_i$ is the estimated methane flowrate and $Q$ is the actual methane flowrate. For each factor being investigated, we grouped the results depending on whether the measurement was an under- or overestimate of the true methane flowrate. We calculated the accuracy of measurements as a range spanning from the median of the over- and

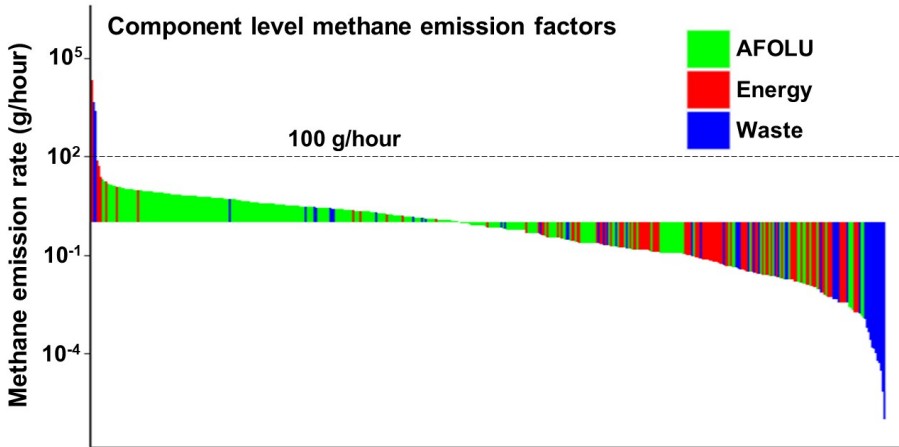

**Figure 2.** Component level methane emission factors from the IPCC emission factor database. Emission factors are categorized according to their respective IPCC source category. All emission factors were converted to methane mass flowrates based on assumptions outlined in the SI-Section 1.1.

underestimated methane flowrates, respectively. We determined the bias of measurements as the average of the raw percentage errors to determine whether tests were biased more towards the under- or overestimation of methane flowrates.

$$E = \frac{Q_i - Q}{Q} * 100 \tag{3}$$

## 3 Results

### 3.1 Prior controlled methane releases and component level methane emissions

We compiled a total of 1,142 component level methane emission factors from the IPCC emission factor database (IPCC-EFDB, 2022). A total of 718 emission factors were from the AFOLU sector, 291 were from the energy sector, and 133 were from the waste sector. The emission factors ranged from $9.8 \times 10^5$ to $-1.1 \times 10^{-2}$ g/hour. We found that 1% of emission factors were above 100 g/hour, 5% of emission factors were above 10 g/hour, and 45% of emission factors were above 1 g/hour. The remaining 55% of emission factors were below 1 g/hour. Within the energy sector the highest component level emission factors were associated with liquid unloadings of storage tanks, flowback events for unconventional oil and gas wells, and fugitive emissions from flaring and venting at oil and gas wells which ranged from $9.8 \times 10^5$ to $1.6 \times 10^5$ g/hour. For the waste sector, the highest component level emission factors were associated with leachate collections wells, pump stations, and sludge pits from landfills which ranged from $4.3 \times 10^3$ to $2.4 \times 10^3$ g/hour. For the AFOLU sector, no component level emission factors were above 100 g/hour, but the highest component level methane emissions we observed from the AFOLU sector were from enteric fermentation from dairy cattle which emitted in the range of 10 g/hour (Figure 2).

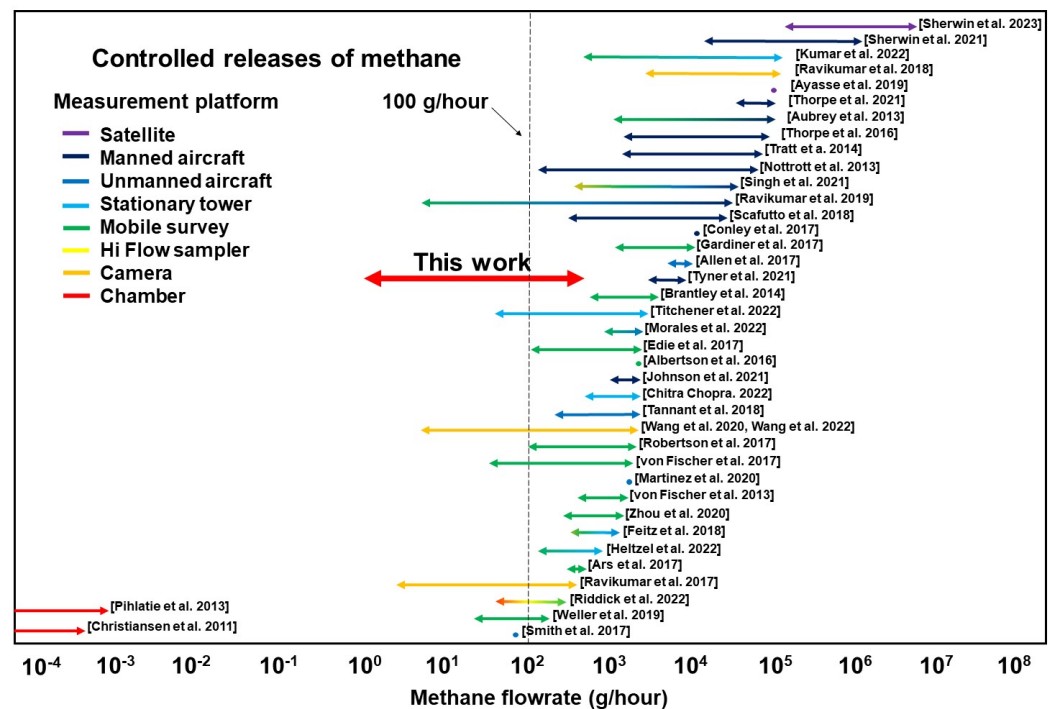

**Figure 3.** Summary of published literature of controlled releases of methane showing the range of methane emissions being tested and coloured according to the measurement platform used to quantify emissions.

We analyzed a total of 40 controlled release studies spanning from 2011 to 2023 (Figure 3). We found that 32 of the 40 (i.e., 80%) controlled release tests had upper methane emission ranges that exceeded 1,000 g/hour, with the highest tested flowrate at

$7.2 \times 10^6$ g/hour for a satellite based platform (Sherwin et al., 2023). We also saw that 31 of the 40 (i.e., 78%) controlled release tests had a lower methane emission range that exceeded 100 g/hour. The majority of controlled releases focused on indirect sampling methods, especially mobile surveying (i.e., 45%) and manned aircraft (i.e., 30%) based measurement platforms (Figure 2). Other indirect methods that were tested less frequently in our review were unmanned aircraft (i.e., 15%), stationary tower (i.e., 13%), and satellite (i.e., 5%) based methods. For direct measurement methods, we observed that camera-based

methods were tested the most frequently (i.e., 10%). We only found three studies that conducted controlled methane releases for chamber based methodologies (Riddick et al., 2022; Pihlatie et al., 2013; Christiansen et al., 2011). We found that eight studies performed controlled releases using multiple measurement methods, with two studies (Singh et al., 2021; Riddick et al., 2022) employing five different methodologies. Overall, we found that the majority of controlled release tests we analyzed focused on indirect sampling methods, and tested methane emission ranges of $\geq$ 100 g/hour. Therefore, the testing we present

here (1.02 g/hour to 512 g/hour) fills this gap and provides guidance for measuring an appropriate range of component level methane sources.

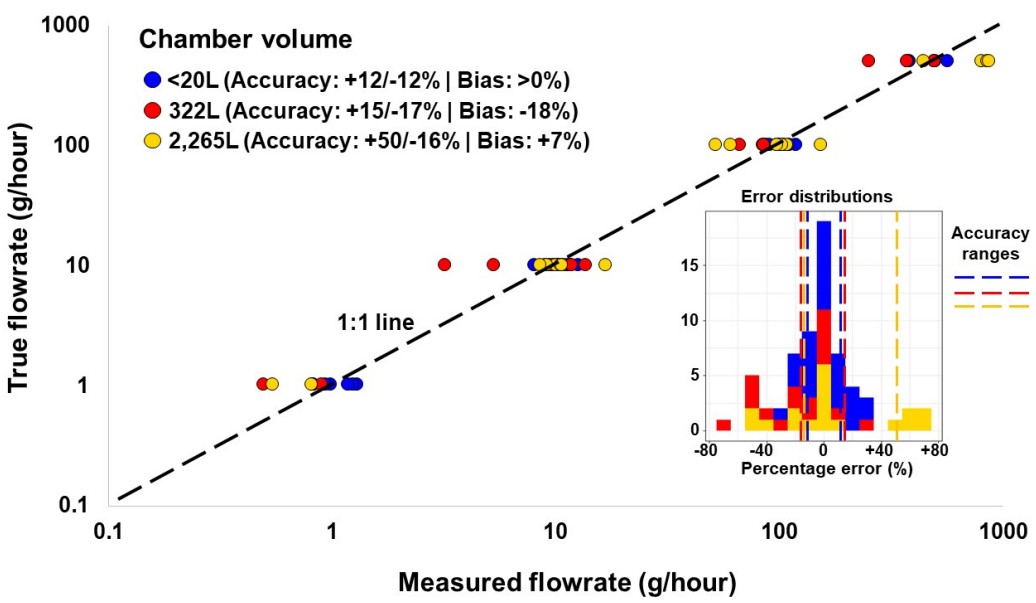

**Figure 4.** Parity plot showing the true versus measured measured methane flowrates for different chamber volumes. The distribution of actual percentage errors is shown in the right. Points and bars are coloured according to the different chamber volumes. The perfect fit line is shown by a dashed black line.

## 3.2 Controlled releases of methane

The accuracy of our 64 controlled release experiments was +14/-14% with a standard deviation of 19%. The average absolute percentage error was ±20% and the median absolute percentage error was ±14%. The lowest error we observed was 0.2%, and 25 of 64 controlled release tests (i.e., 39%) had percentage errors lower than ±10%. Based on testing for bias, we found that the average percentage difference between actual and measured mass flowrates to be -3%, implying a small bias towards the underestimation of methane flowrates.

### 3.2.1 Chamber volume

Our analysis of chamber volume with respect to quantification accuracy showed that the accuracy of measurements increased with smaller chamber volumes (Figure 4). The ≤20 L chambers had the highest accuracy at +12/-12% with an error standard deviation of 12%. The 322 L chamber had a lower accuracy of +15/-17% with a standard deviation of 23%. Our highest errors were measured from the largest 2,265 L chamber with an accuracy of +50/-16% and a standard deviation of 26%. We analyzed all three chamber sizes for bias and found that the ≤20 L chambers showed a slight tendency for underestimation of flowrates with an average bias of ≥0%, the 322 L chamber showed a stronger tendency towards the underestimation of flowrates at -18%, and the 2,265 L chamber showed a slight bias towards overestimating flowrates at +7% (Figure 4).

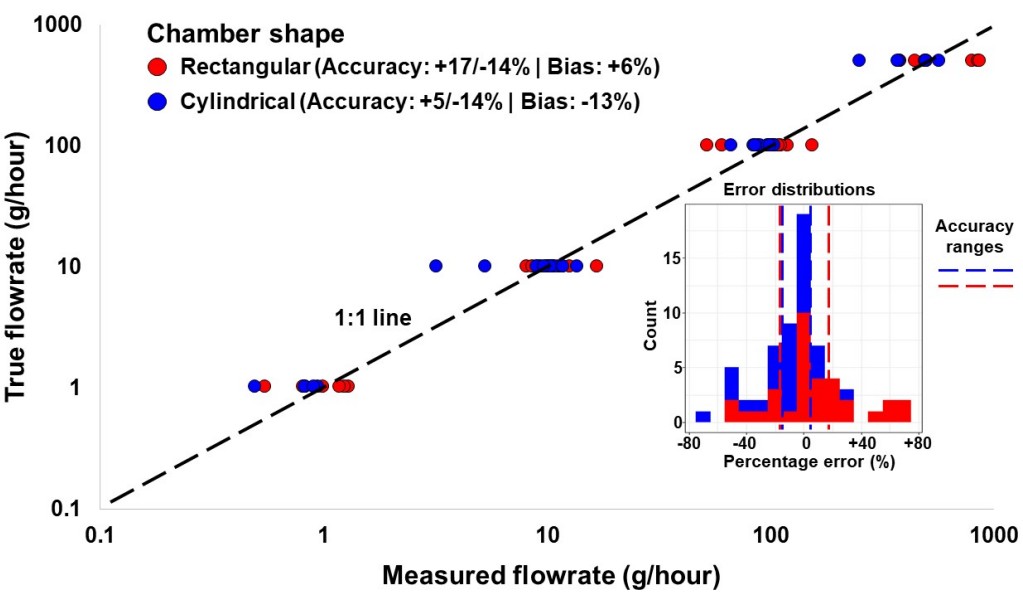

**Figure 5.** Parity plot showing the true versus measured measured methane flowrates for different chamber shapes. The distribution of actual percentage errors is shown in the right. Points and bars are coloured according to the different chamber shapes. The perfect fit line is shown by a dashed black line.

### 3.2.2 Chamber shape

Our comparisons of different chamber shapes showed that the cylindrical chambers were more accurate than the rectangular chambers, showing an accuracy of +5/-14% and a standard deviation of 18% (Figure 5). We found that the rectangular chambers showed a lower accuracy of +17/-14% with a standard deviation of 22%. Similar to the chamber volume, the median percentage error was smaller than the average error for both chamber shapes, which indicates an extreme distribution in percentage errors. We analyzed both chamber shapes for bias and found that the cylindrical chambers were biased towards the underestimation of methane flowrates with an average bias of -13% whereas the rectangular chambers showed a small bias towards the overestimation of methane flowrates with an average bias of +6% (Figure 5).

### 3.2.3 Usage of fans

The most impactful physical factor we observed on chamber measurement accuracy was the presence of fans, where chambers with fans present had a median percentage error of +6/-5% and a standard deviation of 17% (Figure 6), which was higher than chambers without fans which had an accuracy of +17/-17% and a standard deviation of 22%. For both data-sets we observed median values lower than the mean indicating a skewed data-set. We analyzed both data-sets for bias and found that both chambers with and without fans showed slight biases towards the underestimation of methane flowrates at -2% and -4% respectively (Figure 6).

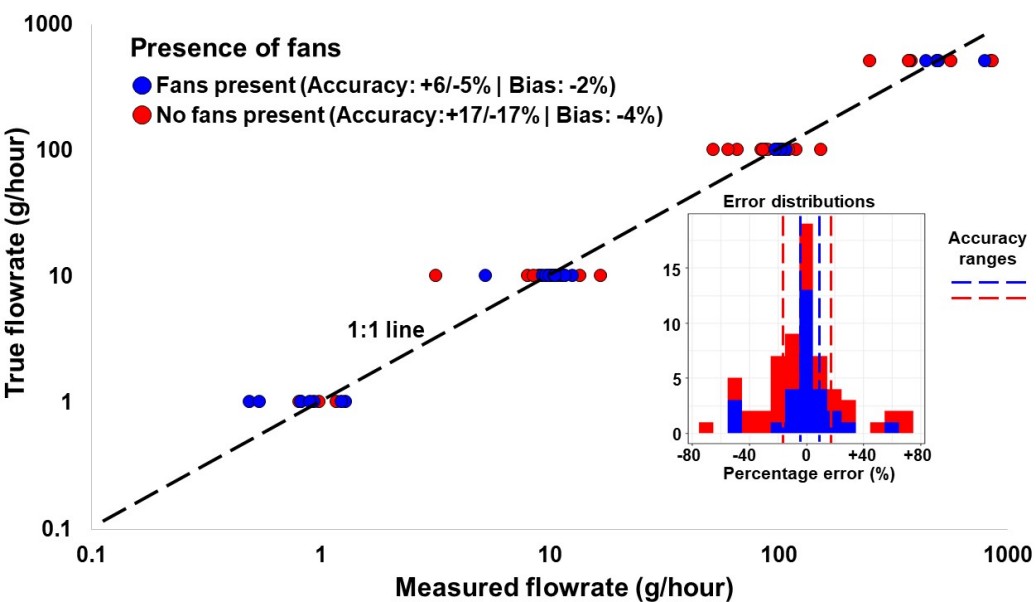

**Figure 6.** Parity plot showing the true versus measured measured methane flowrates for experiments with and without fans present. The distribution of actual percentage errors is shown in the right. Points and bars are coloured according to whether fans were present or not. The perfect fit line is shown by a dashed black line.

## 3.3 Effects of leak properties

### 3.3.1 Mass flowrate

We tested four different mass flowrates for our controlled release tests: 1.02 g/hour, 10.2 g/hour, 102 g/hour, and 511 g/hour (Figure 7). The lowest errors were measured from the 10.2 and 102 g/hour mass flowrates each with accuracies of +8/-11%
and +7/-13% respectively. The lowest accuracy of +56/-15% was attributed to the highest mass flowrate of 512 g/hour. We found that the 1.02, 10.2, and 102 g/hour mass flowrates all had negative biases of -11%, -1%, and -6% respectively. The mass flowrate of 512 g/hour had a slight bias of +4% towards the overestimation of mass flowrates, and also the highest upper accuracy estimate of +46% we observe among the different factors we analyzed.

### 3.3.2 Volumetric flowrate

We analyzed six different volumetric flowrates for the range of methane flowrates we tested: 0.238 L/min, 0.476 L/min, 2.38 L/min, 4.76 L/min, 11.9 L/min, and 23.8 L/min (Figure 7). We found that the lowest accuracies were attributed to both the highest and lowest volumetric flowrates with accuracies of +50/-15% and +21/-14% respectively, whereas higher accuracy was observed with the mid-level volumetric flowrates of 11.8, 4.76, 2.38, and 0.476 SLPM with accuracies ranging from +26/-3%

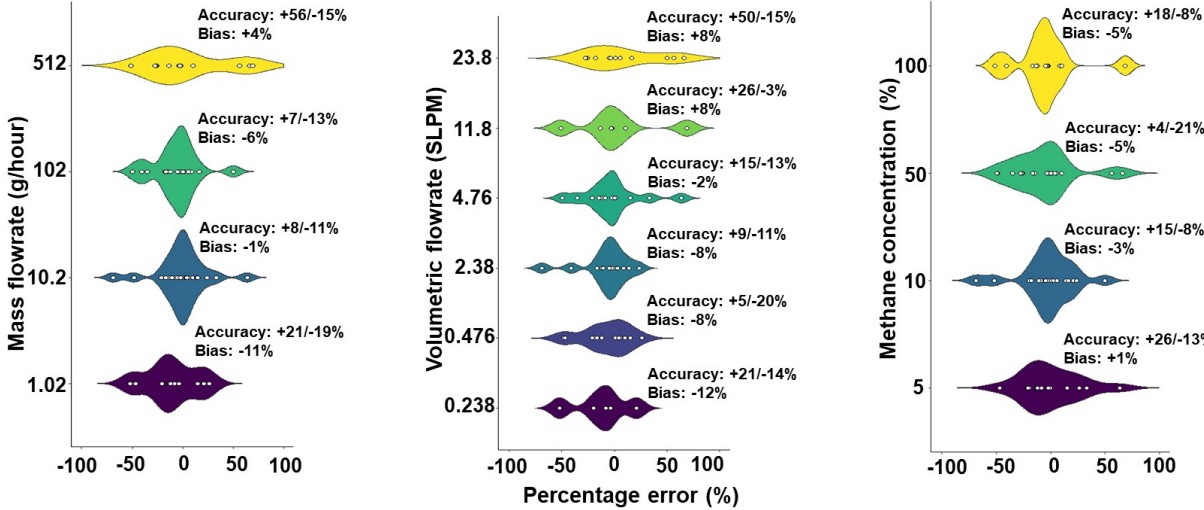

**Figure 7.** Violin plots of the percentage errors of true versus measured methane flowrates under varying mass flowrates (left), volumetric flowrates (middle), and gas concentrations (right) of methane. The points represent the measured percentage errors, and the shaded areas represent the relative density (on the y-axis) of the observed percentage errors. Uncertainty ranges and biases are displayed for each factor.

to +9/-11%. Similar to the mass flowrates, we also found the highest accuracies were associated with the mid-level volumetric
flowrates while the lowest accuracies were observed at the upper and lower volumetric flowrates.

### 3.3.3   Methane percentage of leaking gas

We analyzed four different percentages of methane in the leaking gas for the controlled releases (Figure 7). The lowest accuracies were associated with the 5% methane gas with an accuracy of +31/-16%, whereas the highest accuracies were observed with the 10% methane at +15/-8%. The three highest percentages of methane in the leaking gas all had small negative biases
ranging from -5% to -3%, whereas the 5% methane leak had a slight positive bias at +1%.

### 3.4   Optimizing the static chamber method for accuracy

For consistency, we define release rates of 1.02 and 10.2 g/hour as small flowrates, and releases of 102 and 512 g/hour as high flowrates. In addition, we define chamber volumes $\leq$20 L as small, and chamber volumes of 322 L and 2,265 L as large. We analyzed how chamber configurations (i.e., chamber volume, usage of fans, chamber shapes) can be optimized to increase
the accuracy of methane flowrate estimates. In general, we found that smaller sized chambers produced the lowest errors. No measurements from a smaller sized chamber produced a percentage error above $\pm$30%. We also saw that smaller chambers performed similarly if fans were present or not, with smaller chambers with fans producing an accuracy of +16/-8% and smaller chambers without fans having an accuracy of +12/-13%. Smaller chambers performed slightly better if the chambers were cylindrical, with an accuracy of +3/-12% compared to smaller rectangular chambers that had an accuracy of +16/-3%.

For larger sized chambers (i.e., $\geq$20L), the usage of fans was critical for reducing measurement error. Larger chambers with fans produced an accuracy of +4/-5% compared to large chambers without fans which produced an accuracy of +63/-27%. Therefore, although smaller chambers generally have lower errors than larger chambers, the errors in the larger chambers can be comparable to the smaller chambers when fans are used.

We found that chamber configurations could also be optimized according to the mass flowrate of methane. At low mass flowrates of methane (i.e., $\leq$ 100 g/hour), we found that smaller sized chambers were more accurate than larger chambers with accuracies of +12/-8% and +15/-19% respectively. The usage of fans had little impact on the accuracy of smaller sized chambers at these low flowrates, with smaller chambers with fans producing an accuracy of +16/-8% and smaller chambers without fans having an accuracy of +7/-13%. In contrast, the usage of fans was important for the accuracy of larger chambers at these lower mass flowrates. Larger chambers with fans had an accuracy of +4/-30% and larger chambers without fans had an accuracy of +48/-19%. In terms of chamber shape, at low flowrates smaller cylindrical chambers had an accuracy of +1/-11% compared to small rectangular chambers which produced an accuracy of +15/-3%. For larger chambers at low mass flowrates, we observed a contrasting result with large rectangular chambers producing an accuracy of +6/-16% and large cylindrical chambers producing a median percentage error of +24/-48%.

We observed similar results for optimizing chamber configurations for high methane mass flowrates (i.e., $\geq$100 g/hour). We found that smaller chambers ($\leq$20 L) performed better than larger chambers with accuracies of +14/-13% and +50/-16% respectively. We found that the usage of fans was critical for measurement accuracy for larger sized chambers at these higher mass flowrates of methane. Larger chambers with fans had an accuracy of +4/-4% compared to larger chambers without fans which had an accuracy of +66/-35%. For chamber shapes, cylindrical chambers were more accurate than rectangular chambers with an accuracy of +6/-14% compared to +26/-15% from rectangular chambers. At higher mass flowrates of methane, we found that large cylindrical chambers with fans were highly accurate at +2/-3% of the true methane flowrate.

From all of the controlled release experiments we performed, we saw that the median absolute error of $\pm$14% was lower than the mean error of 20%, indicating a heavy-tailed distribution of measurement errors. As such, we analyzed all controlled release experiments where the resulting error exceeded 40% to assess the potential cause of these erroneous measurements. A total of 12 controlled releases had quantification errors that exceeded 40% (SI - Figure S1). All of these experiments were conducted on larger volume chambers (i.e., 322L and 2,265L), and 8 of the 12 had no fans present. Based on a comparison of the fit of the linear regressions, we found that these 12 experiments did have good correlation between methane concentrations and time with $R^2$ values averaging 0.91 when compared to the rest of the data-set (mean $R^2$ = 0.96). Notably, 3 of the 12 high-error measurements had very high $R^2$ values exceeding 0.99, with an example being shown in the bottom left of Figure 8. We observed a similar phenomena with a single controlled release performed with an "ideal" and "non-ideal" chamber seal which is shown in the SI - Section 1, where a "non-ideal" chamber seal produced a high $R^2$ value yet underestimated the methane flowrate by 43%.

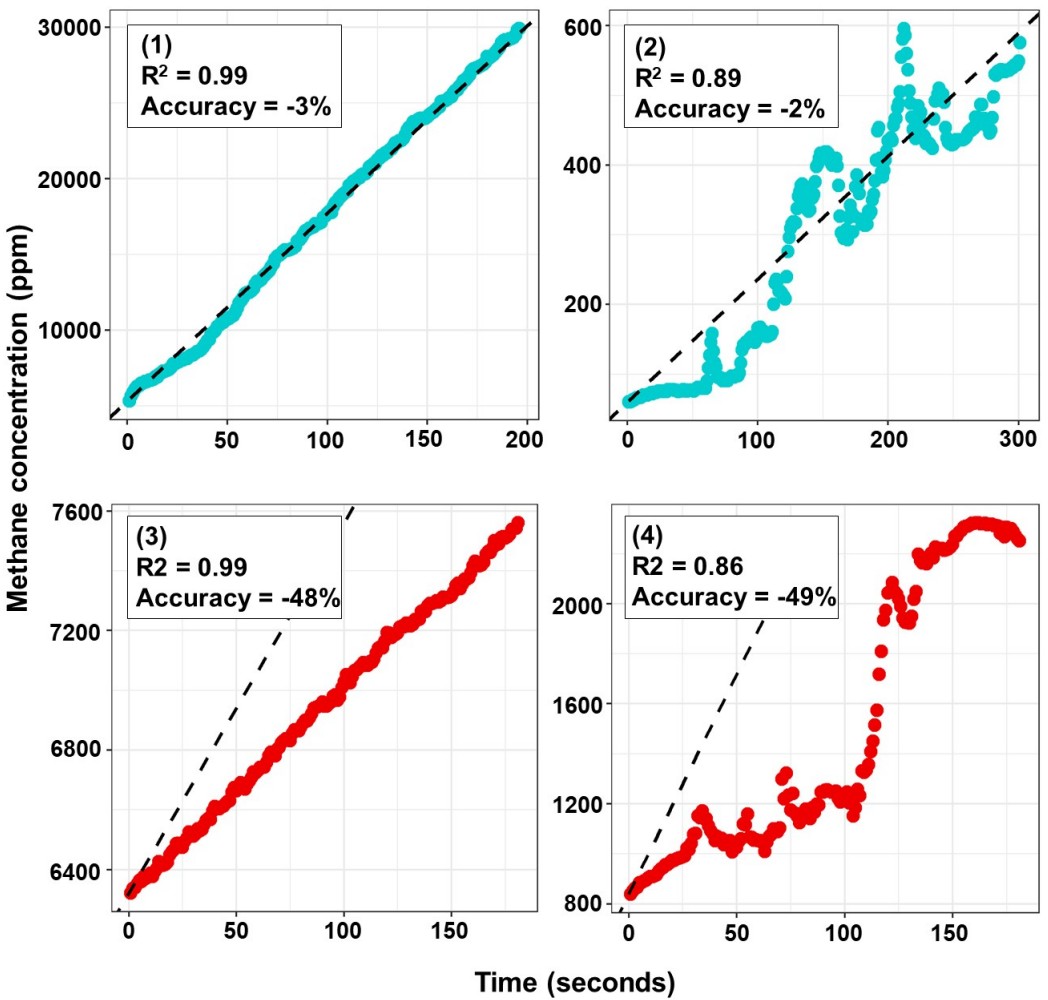

**Figure 8.** Four examples of raw controlled release data showing methane concentrations versus time. The measured concentrations within the chamber are shown by the coloured dots, and the true or expected concentrations are indicated by the dashed lines. Points coloured blue indicate a measurement error less than 40%, and the points coloured red indicate a measurement error greater than 40%. The examples shown are 1) high $R^2$ value and high quantification accuracy; 2) low $R^2$ value and high quantification accuracy; 3) high $R^2$ value and low quantification accuracy; 4) low $R^2$ value and low quantification accuracy.

## 4 Discussion

Our compilation of component level methane flowrates from the IPCC emission factor database showed that 99% of the component level emission rates fall below the 100 g/hour level. Therefore, it is important to develop and test methane quantification methods for these lower methane flowrates (i.e., $\leq 100$ g/hour). Quantification of methane emissions at the component level

provides a level of detail necessary to develop actionable mitigation strategies through the clear identification of emitting components. Most controlled release studies focus on indirect sampling methods which are effective in measuring methane emissions at the site and/or facility level scale. While these data are important for validating greenhouse gas inventories and quantifying emissions from super-emitting methane sources (Brandt et al., 2016; Ravikumar et al., 2017), emissions data at the component level are also needed to improve bottom-up greenhouse gas inventories and develop actionable mitigation strategies. Many of the component level sources we consider such as manholes, livestock, abandoned oil and gas wells, and NG pipeline leaks have all been shown to be significant methane sources at municipal, provincial/state/territorial, and national levels (Williams et al., 2022, 2020; El Hachem et al., 2022; Seiler et al., 1983; Kang et al., 2016; Hendrick et al., 2016). These sources are all characterized by low methane emissions rates below 100 g/hour range on average, which are challenging to measure using indirect methods. Several studies have highlighted the super-emitting nature of methane emission sources, particularly from the NG sector (Brandt et al., 2016). However, the upper range of super-emitting methane sources varies depending on the source being measured. For example, a study of methane emissions from Montreal, Canada, found that both residential NG meter-sets and manholes were significant sources of methane for the city despite having maximum methane emission rates of 4.2 and 33 g/hour respectively (Williams et al., 2022). While many controlled release studies focus on a higher range of methane emissions, it is still important that methods are developed and tested for lower methane emitting sources.

In addition to the factors we tested, there are several other sources of uncertainty in the static chamber method that we did not investigate. One factor that could impact measurement accuracy is the effectiveness of the chamber seal. An improper chamber seal could lead to intrusion from atmospheric air which dilutes the chamber headspace leading to an underestimation of the true methane flow rate (SI - Section 1). Typically in field settings, chambers are sealed to the ground (Kang et al., 2016; Lebel et al., 2020). In some cases, chambers can be sealed above-ground to an emitting component (Figure 1). A variety of different methods have been used to create these chamber-to-site seals, such as tape, bungie chords, chamber collars, sand, snow, etc (Williams et al., 2020; Lebel et al., 2020; Kang et al., 2016). In our experience, smaller chambers are easier to seal to an emitting component given the smaller size and ease in identifying potential breaches. Ensuring a proper chamber seal in larger chambers is more difficult due to the chamber size but the seal is achievable under stable environmental conditions. The methane concentration measurement method is one aspect of the static chamber method that will affect both the measurement accuracy and sensitivity of the static chamber method. In this work we use a portable greenhouse gas analyzer to continuously measure methane concentrations within the chamber. Uncertainty related to the frequency of methane concentration measurement and the accuracy and precision of the greenhouse gas analyzer are all important factors related to uncertainty. Furthermore, portable greenhouse gas analyzers can generally be classified as either measuring a full range of methane concentrations at the cost of precision at lower methane concentrations (i.e., $\leq 10$ ppm methane), or measuring methane with high precision at the cost of an upper measurement range (i.e., 1,000 ppm). Therefore, the selection of the greenhouse analyzer can also be optimized according to the methane source being measured to improve accuracy. Other factors such as the release point of the emitted gas; presence of multiple emission sources; environment; chamber rigidity; method and strength of interior chamber mixing; and position of the gas sampling points are all factors that could also impact measurement uncertainty. Further analysis of

the impacts of these factors on measurement accuracy would be beneficial for guiding ideal deployment of the static chamber method for the quantification of component-level methane sources.

Our results showed that the static chamber methodology can quantify methane emissions ranging from 1.02 g/hour to 512 g/hour with an accuracy of +14/-14%. In comparison to indirect methods, Johnson et al. (2023) state that their aircraft-based method has a multi-pass uncertainty range of -46/+54%, which roughly corresponds to an absolute error of ±50%. In von Fischer et al. (2017), they state an uncertainty range of -24/+32% after five mobile survey passes, which roughly corresponds to an absolute error of ±28%. With regards to other controlled release tests on static chambers, we do find that our median uncertainty of +14/-14% falls within the 10-20% range reported by Lebel et al. (2020) and Pihlatie et al. (2013). For the larger chambers, we found that the usage of fans was critical for maximizing accuracy, which is expected given the larger volume of air that is required to be mixed. All of the 12 largest measurement errors occurred from large volume chambers, with 8 of those controlled releases having no fans present. The larger chambers we used were all collapsible chambers, which could have impacted measurement accuracy through pressure pumping in the chamber headspace through wind impacting the collapsible chamber walls and altering the chamber volumes throughout the experiment. The large volume chambers we used are designed to accommodate odd site shapes encountered in the field, such as abandoned oil and gas wells (Figure 1). Future controlled release studies that test larger volume rigid chambers would help elucidate the cause of these high errors. We also noted that all of these large measurement error experiments showed high $R^2$ values above 0.80, meaning that they would be difficult to distinguish based on the goodness-of-fit of the measurement data alone. Furthermore, several experiments showed relatively poor $R^2$ values but good measurement accuracy (Figure 8), adding to this difficulty. We found that chamber shape is more important for larger chambers than smaller chambers, with the large cylindrical chamber performing better than the large rectangular chamber whereas we did not find any difference between the smaller sized chambers with respect to shape. Ideally static chambers should be constructed to minimize potential "dead zones" where gases can accumulate (Christiansen et al., 2011), and cylindrical, or even semi-spherical or spherical chambers, should facilitate easier mixing of the chamber headspace.

At higher methane flowrates (≥100 g/hour) we found that our large cylindrical chamber with fans quantified methane emissions with the highest accuracy (i.e., +2/-3%) of any chamber combination we used throughout this study. In addition, a methane source such as an oil and gas well can have multiple emitting components (e.g., pipe flanges, valves, surface casing vents, soil gas migration) which could be missed if using smaller sized chambers. Methane concentrations within a smaller chamber can also rapidly reach explosive levels which can pose safety concerns if the environment is not intrinsically safe (Riddick et al., 2022), but these risks can be minimized at little cost to accuracy if fans are omitted. Furthermore, intrinsically safe methods of chamber mixing such as external pumps could be used to mix air within chambers, regardless of the size of chamber. Theoretically, there is no upper methane flowrate limitation of the static chamber method, and utilizing large chambers such as the 32,000 L chamber used in Lebel et al. (2020) could theoretically quantify methane flowrates in the 100-200 kg/hour range. However, there are practical limitations to directly measuring components emitting methane at these high levels, the most notable being safety concerns and access issues (e.g., measuring flare stacks and liquid storage tank unloadings). Another factor to consider is the time to reach steady-state. Enclosing a high methane emitting source within a smaller chamber causes methane concentrations within the chamber to rapidly reach steady-state, essentially creating a

350  dynamic chamber, which we do not test in this work (Pedersen et al., 2010; Levy et al., 2011). Overall, our findings indicate that small chambers (i.e., $\leq$20 L), regardless of the chamber shape and usage of fans, can be used to quantify component level methane flowrates with an accuracy of $\pm 11\%$ for methane flowrates ranging from 1.02 to 512 g/hour. If larger chambers are required/desired, optimal configurations (i.e., fans present and cylindrical shapes) will produce errors $\pm 3\%$ for high methane flowrates (i.e., $\geq$100 g/hour).

## 5  Conclusions

Our results have shown that the static chamber methodology can be an effective and accurate method for the quantification of component level methane flowrates. While indirect sampling methods have been tested extensively, there is a need to test direct sampling methods given their ability to quantify methane emissions at the component level, which is important for developing actionable mitigation strategies. The static chamber method is logistically simple to implement and adaptable to multiple methane sources, making it a viable measurement option for many component level emission sources. Going forward, there are opportunities to improve the static chamber design to reduce measurement uncertainties. Our work provides the testing and design information for the static chamber methodology, thereby contributing to the range of measurement tools needed to quantify methane emission rates from all sources.

*Author contributions.*  JPW wrote and edited this manuscript, including the data analysis and development of figures and tables. JPW, MK, and KEH all contributed to the development of the controlled release testing measurement plan and the editing of the manuscript. JPW and KEH performed the controlled release experiments.

*Competing interests.*  The authors declare no competing interests

*Data availability.*  Excel files of the IPCC emission factor database compilation are available for the AFOLU, Energy, and Waste sectors. Conversions to component level emission factors and justifications are provided within each excel file.

*Acknowledgements.*  We would like to acknowledge the members of the Kang Lab at McGill University for their valuable insights in organizing, performing, and summarizing the work in this manuscript.

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
