# Peer review of "Controlled release testing of the static chamber methodology for direct measurements of methane emissions"

_Atmospheric Measurement Techniques, 2023_

## Author Comment (AC1)

We would like to thank all reviewers for their valuable insights and comments made to improve this work. We believe we have addressed all comments made by the reviewers as shown below, and the suggested changes have greatly improved this manuscript.

Our response to all comments is structured so that the reviewers comments are shown in "**bold**", our responses are shown in *"italics"*, prior text in the manuscript is shown in "blue", added text is shown in "red", and any deleted text in the manuscript is shown in "blue strikethrough".

We would once again like to thank the reviewers for their valuable insights and comments regarding this manuscript.

In addition to the changes outlined below for this reviewer, we also performed additional edits that we believe improve on the current version of the manuscript without significantly altering any results. First, we address some previously unmentioned methods we used for the linear regressions for controlled releases where methane concentrations were expected to rapidly reach steady-state. We also modified our presentation of the accuracy of measurements to encapsulate the median of the over- and underestimates to better represent the bias in our measurements. Finally, we incorporated two new studies on prior controlled releases into Figure 2 and the subsequent analysis section. The changes are outlined as follows:

Line 140: "For some experiments, methane concentrations within the chamber were expected to rapidly reach steady-state. Steady-state is reached when methane concentrations no longer increase over time in the chamber and the concentration of methane within the chamber is equal to the concentration of the released gas. The residence time, or time to reach steady-state, is defined by (2): where $\tau$ is the residence time, $V$ is the volume of the chamber, and $Q$ is the volumetric flowrate of gas (i.e., methane and balance gas combined) into the chamber. For any controlled releases where the expected residence time was two minutes or lower, we only used the initial ten data points for the linear regression to avoid the period of exponential decay as methane concentrations approach steady-state (Pihlatie et al., 2013).

(3) $\tau = V/Q$

For each factor being investigated, we grouped the results depending on whether the measurement was an under- or overestimate of the true methane flowrate. We calculated the accuracy of measurements as a range spanning from the median of the over- and underestimated methane flowrates, respectively. We determined the bias of measurements as the average of the raw percentage errors to determine whether tests were biased more towards the under- or overestimation of methane flowrates. "

New Figure 2:

[Figure]

**Review #1**

**I enjoyed reading this manuscript. Generally, the manuscript is well written, identifies the knowledge gap clearly on missing information on chamber-based measurements and performs well-documented research to add new knowledge that is highly applicable.**

**Language is also good, but it becomes rather monotonous in the Results section to read over and over again the same phrase "We observed..." etc. Not that this choice of words disqualifies the results, but the authors could think of varying the language in this respect. It is a minor thing and totally up to the authors. Just a point of view from the reader.**

*We agree with the reviewer and have edited text for improved flow for the reader, especially in the Results section where the consistent use of "We found" and "We observed" becomes monotonous as stated. Any changes to improve the experience of the reader are important, so we welcome any comments in that regard. We outline the changes below:*

Line 178-185: "Our analysis of chamber volume with respect to quantification accuracy showed that the accuracy of measurements increased with smaller chamber volumes (Figure 4). The <20 L chambers had the highest accuracy at +12/-12% with an error standard deviation of 12%. The 322 L chamber had a lower accuracy of +15/-17% with a standard deviation of 23%. Our highest errors were measured from the largest 2,265 L chamber with an accuracy of +50/-16% and a standard deviation of 26%. We analyzed all three chamber sizes for bias and found that the <20 L chambers showed a slight tendency for underestimation of flowrates with an average bias of >0%, the 322 L chamber showed a stronger tendency towards the underestimation of flowrates at -18%, and the 2,265 L chamber showed a slight bias towards overestimating flowrates at +7% (Figure 4)."

Line 187-193: "Our comparisons of different chamber shapes showed that the cylindrical chambers were more accurate than the rectangular chambers, showing an accuracy of +5/-14% and a standard deviation of 18% (Figure 5). We found that the rectangular chambers showed a lower accuracy of +17/-14% with a standard deviation of 22%. Similar to the chamber volume, the median percentage error was smaller than the average error for both chamber shapes, which indicates an extreme distribution in percentage errors. We analyzed both chamber shapes for bias and found that the cylindrical chambers were biased towards the underestimation of methane flowrates with an average bias of -13% whereas the rectangular chambers showed a small bias towards the overestimation of methane flowrates with an average bias of +6% (Figure 5)."

Line 195-199: "The most impactful physical factor we observed on chamber measurement accuracy was the presence of fans, where chambers with fans present had a median percentage error of +6/-5% and a standard deviation of 17% (Figure 6), which was higher than chambers without fans which had an accuracy of +17/-17% and a standard deviation of 22%. For both data-sets we observed median values lower than the mean indicating a skewed data-set. We analyzed both data-sets for bias and found that both chambers with and without fans showed slight biases towards the underestimation of methane flowrates at -2% and -4% respectively (Figure 6)."

Line 202-208: "We tested four different mass flowrates for our controlled release tests: 1.02 g/hour, 10.2 g/hour, 102 g/hour, and 511 g/hour (Figure 7). The lowest errors were measured from the 10.2 and 102 g/hour mass flowrates each with accuracies of +8/-11% and +7/-13% respectively. The lowest accuracy of +56/-15% was attributed to the highest mass flowrate of 512 g/hour. We found that the 1.02, 10.2, and 102 g/hour mass flowrates all had negative biases of -11%, -1%, and -6% respectively. The mass flowrate of 512 g/hour had a slight bias of +4% towards the overestimation of mass

flowrates, and also the highest upper accuracy estimate of +46% we observe among the different factors we analyzed."

Line 210-215: "We analyzed six different volumetric flowrates for the range of methane flowrates we tested: 0.238 L/min, 0.476 L/min, 2.38 L/min, 4.76 L/min, 11.9 L/min, and 23.8 L/min (Figure 7). We found that the lowest accuracies were attributed to both the highest and lowest volumetric flowrates with accuracies of +50/-15% and +21/-14% respectively, whereas higher accuracy was observed with the mid-level volumetric flowrates of 11.8, 4.76, 2.38, and 0.476 SLPM with accuracies ranging from +26/-3% to +9/-11%. Similar to the mass flowrates, we also found the highest accuracies were associated with the mid-level volumetric flowrates while the lowest accuracies were observed at the upper and lower volumetric flowrates."

Line 217-222: "We analyzed four different percentages of methane in the leaking gas for the controlled releases (Figure 7). The lowest accuracies were associated with the 5% methane gas with an accuracy of +31/-16%, whereas the highest accuracies were observed with the 10% methane at +15/-8%. The three highest percentages of methane in the leaking gas all had small negative biases ranging from -5% to -3%, whereas the 5% methane leak had a slight positive bias at +1%."

Line 225-226: "We  analyzed how  chamber configurations (i.e., chamber volume, usage of fans, chamber shapes) can be optimized…"

Line 229: " We also  saw that smaller chambers performed similarly if fans…"

Line 231: "Smaller chambers performed slightly better…"

Line 233: "For larger sized chambers (i.e., ≥20L),  the usage of fans was critical…"

Line 238-250: "At low mass flowrates of methane (i.e., ≤ 100 g/hour), we found that smaller sized chambers were more accurate than larger chambers, with  with accuracies of +12/-8% and +15/-19% respectively. The usage of fans had little impact on the accuracy of smaller sized chambers at these low flowrates, with smaller chambers with fans producing  an accuracy of +16/-8% and smaller chambers without fans having an accuracy of +7/-13% . In contrast,  the usage of fans was important for the accuracy of larger chambers at  lower mass flowrates Larger chambers with fans  had an accuracy of +4/-30% and larger chambers without fans had an accuracy of +48/-19%. In terms of chamber shape,

 at low flowrates smaller cylindrical chambers had an accuracy of +1/-11% compared to small rectangular chambers which produced an accuracy of +15/-3%  For larger chambers at low mass flowrates, we  observed a contrasting result with large rectangular chambers producing an accuracy of +6/-16% and large cylindrical chambers producing a median percentage error of +24/-48%.

Line 251-258: "We observed similar results for optimizing chamber configurations for high methane mass flowrates (i.e., ≥100 g/hour). We found  that the usage of fans was critical for measurement accuracy for larger sized chambers at these higher mass flowrates of methane. Larger chambers with fans had an accuracy of +4/-4%  compared to larger chambers without fans which had had an accuracy of +66/-35% . For chamber shapes,  cylindrical chambers were more accurate than rectangular chambers with an accuracy of +6/-14% compared to +26/-15% from rectangular chambers . At  higher mass flowrates of methane, we found that large cylindrical chambers  were highly accurate at +2/-3% of the true methane flowrate . "

**Figure quality can be improved and below I suggest adding more figures on the actual chamber designs and chamber measurements to make it more tangible for the reader.**

*We agree with the reviewer and have modified all figures in this work. Figure 1 was modified to include two controlled release tests that were not included in the original submission. Figure 2 was modified with small aesthetic changes. Figure 3 was changed from violin plots to three scatter plots showing measured versus true methane flowrates, with the tested factors highlighted through the colour/size/shape of the points. We have also added a new Figure 1 to the manuscript that shows the general experimental design as well as pictures of all the tested chambers in either field settings or the controlled release experiment itself. We believe that having pictures of the chambers deployed in field settings adds additional information to the reader and gives good reference points for later discussion, especially for questions relating to chamber seal in-situ. We also deleted Figure 5 and replaced it with a figure showing four examples of raw data from the controlled releases.*

[Figure]

*New Figure 1*

*Example of replacement figure for Figure 3*

[Figure]

*New Figure 8 that replaces old Figure 5*

I have marked "reconsidered after major revisions" although the major work is not revising the text for flaws per se, but more adding new figures and text to increase the value of the manuscript. I hope you can see my point here as outlined below.

**Materials and Methods**

Your description of the experiment is clear, but I was missing pictures of your chambers in the field. This can provide valuable information for the reader wanting to do the same as you, but also help the reader/reviewer assess critical design aspects of your chambers that are not apparent, directly from your description in text. Also, since the chamber testing is the central element in your paper it deserves to be highlighted. Therefore, I suggest to make a new Figure 1 with four panels each showing the chambers A – D. Here it will also be obvious to show the methane delivery system and other important details.

*We agree with your comments and have added a new Figure 1, as described earlier in this response. We have also added a new figure that shows the results of a subset of controlled release experiments that highlight a range of measurement accuracies and goodness-of-fits.*

**Another thing I was missing was the description of how the chambers are sealed to the ground? Sealing of the chamber to the ground is an essential part of the chamber based measurement and is critical when measuring under windy conditions. Even a lower windspeeds between 5 – 15 kph, as you report, you can have a relatively large disturbance of the chamber headspace concentration that will negatively impact your flux calculation, by leading to underestimation if chamber headspace is diluted which could be a likely scenario in your case with very high CH4 concentrations. Please add a description of this in text and this will also be aided by having the pictures of the chambers as suggested.**

*We have added additional descriptions in the text to clarify how chambers were sealed to the ground during these experiments and also new text in the Discussion regarding how chambers are sealed to the ground in field settings, as this process can differ depending on the environment. For example, in some cases chambers are not sealed to the ground at all and are instead sealed around an emitting component (see new Figure 1 below showing chamber C installed over an emitting gas well at a historic landfill site). For our experiments, chambers were sealed to a flat piece of plastic with duct tape, which we believe would constitute a near-ideal chamber seal. We did not specifically test for variations in chamber seals (e.g., using snow or sand to "seal" the chamber to the ground), but we do now have text in the discussion that highlights how impactful this factor could be in measurement accuracy, and how future studies could investigate this further for these component level measurements. We have also added a short paragraph and description in the SI (Section 1) of one prior controlled release test we performed in 2019 which did quantify the effects of an "improper" versus "proper" chamber seal on the larger 2,265L chamber (Chamber A). The resulting difference was a drop in accuracy by 26%. We have included a reference to this in the text.*

Line 279: "One factor that could impact measurement accuracy that we do not investigate is the effectiveness of the chamber seal. An improper chamber seal can lead to intrusion from atmospheric air which dilutes the chamber headspace leading to an underestimation of the true methane flow rate (SI - Section 1). Typically in field settings, chambers are sealed to the ground (Kang et al., 2016, Lebel et al., 2020). IN some cases, chambers can be sealed above-ground to an emitting component (Figure 1). A variety of different methods have been used to create these chamber-to-site seals, such as tape, bungie chords, chamber collars, sand, snow, etc (Lebel et al., 2020, Williams et al., 2022, Kang et al., 2016). In our experience, smaller chambers are easier to seal to an emitting component given the smaller size and ease in identifying

potential breaches. Ensuring a proper chamber seal in larger chambers is more difficult due to the chamber size but the seal is achievable under stable environmental conditions. The  method of analyzing methane concentrations within the chamber is one aspect of the static chamber method that will affect both the measurement accuracy and sensitivity of the method."

Line 288: "Further analysis of the impacts of these factors on measurement accuracy would be beneficial for guiding ideal deployment of the static chamber method for the quantification of component-level methane sources."

[Figure]

**For others that want to measure CH4 fluxes at the component level it becomes quite relevant to know your reflections on the sealing. Also, as I could imagine that surfaces are not always smooth and flat, but can constitute more complex 3D structures?**

*Yes you are correct, in some cases the chambers are fastened above-ground to the emitting components. We hope that the changes outlined in an earlier comment and the new Figure 1 will provide a better explanation of these "new" applications for the static chamber method.*

**Results**

**Short and to the point and I do like the use of the error, so the results become comparable across chamber types, flow rates, with/without fans.**

**The violin plots are good to show the overall performance of the tests, but I was missing some more detailed plots on how actual chamber measurements (CH4 concentration vs time) looked in case with good agreement (~1% error) and poor (>30% error) for both the small and large chambers under calm and windy conditions. In the main manuscript I would just highlight some examples and then in supplementary materials provide all 64 chamber enclosure measurements. In such a figure you could add the theoretical CH4 concentration (e.g. what is dictated according to the release rate) and the actual observed concentration.**

This is an excellent suggestion and we have adopted all of the suggested changes to the main manuscript and the SI. The changes are summarized as:

Line 258: "From all of the controlled release experiments we performed, we saw that the median error of +/-14% was lower than the mean error of 20%, indicating a heavy-tailed distribution of measurement errors. As such, we analyzed all controlled release experiments where the resulting error exceeded 40% to assess the potential cause of these erroneous measurements. A total of 12 controlled releases had quantification errors that exceeded 40% (SI - Figure S1). All of these experiments were conducted on larger volume chambers (i.e., 322L and 2,265L), and 8 of the 12 had no fans present. Based on a comparison of the fit of the linear regressions, we found that these 12 experiments did have lower $R^2$ values averaging 0.91 when compared to the rest of the dataset (mean $R^2$ = 0.96). Notably, 3 of the 12 high-error measurements had very high $R^2$ values exceeding 0.99, with an example being shown in Figure 8. We observed a similar phenomena with a single controlled release performed with an "ideal" and "non-ideal" chamber seal which is shown in the SI - Section 1, where a the "non-ideal" chamber seal produced a high $R^2$ value yet underestimated the methane flowrate by 43%. "

Line 292: "For the larger chambers we  found that the usage of fans  was critical for maximizing accuracy, which is expected given the larger volume of air that is required to be mixed. All of the 12 largest measurement errors occurred from large volume chambers, with 8 of those controlled releases having no fans present. The larger chambers we used were all collapsible chambers, which could have impacted measurement accuracy through pressure pumping in the chamber headspace through wind impacting the collapsible chamber walls and altering the chamber volumes throughout the experiment. The large volume chambers we used were designed to accommodate odd site shapes encountered in the field, such as abandoned oil and gas wells (Figure 1). Future controlled release studies that test larger volume rigid chambers would help elucidate the cause of these high errors. We also noted that all of these large measurement error experiments showed high $R^2$ values above 0.80,

meaning that they would be difficult to distinguish based on the goodness-of-fit of the measurement data alone. Furthermore, several experiments showed relatively poor R2 values but good measurement accuracy (Figure 8), adding to this difficulty. We  find that chamber shape is more important for larger chambers than smaller chambers, with the large cylindrical chamber performing better than the large rectangular chamber, whereas we did not find any difference between the smaller sized chambers with respect to shape. Ideally static chambers should be constructed to minimize potential "dead zones" where gases can accumulate (Christiansen et al., 2011), and cylindrical, or even spherical chambers, should facilitate easier mixing of  the chamber headspace."

Line 300-303: ""Methane concentrations within a smaller chamber can also rapidly reach explosive levels which can pose safety concerns if the environment is not intrinsically safe (Riddick et al., 2022), but these risks can be minimized at little cost to accuracy if fans are omitted. Furthermore, intrinsically safe methods of chamber mixing such as external pumps could be used to mix air within chambers, regardless of the size of chamber. Theoretically, there is no upper methane flowrate limitation of the static chamber method, and utilizing large chambers such as the ~32,000 L chamber used in (Lebel et al., 2020) could theoretically quantify methane flowrates in the 100-200 kg/hour range. However, there are practical limitations to directly measuring components emitting methane at these high levels, the most notable being safety concerns and access issues (e.g., measuring flare stacks and liquid storage tank unloadings). Another factor to consider is the time to reach steady-state. Enclosing a high methane emitting source within a smaller chamber causes methane concentrations within the chamber to rapidly reach steady-state, essentially creating a dynamic chamber, which we do not test in this work (Pedersen et al., 2010, Levy et al., 2011)."

**Discussion**

**Again, well written and to the point and you summarize the main findings well. However, I think some more in depth technical discussion on chamber performance and simulation of methane release is needed to put the value of your study in perspective.**

**I was surprised to see the extreme variation in error for your experiments which seem to be present no matter the combination of mass flow rates, volumetric flowrate and chamber design. You do not discuss this issue, but only use the median error. I think this is a mistake and you should discuss in more depth the reasons why you can have such large errors in some case. Here showing the**

**individual enclosures to the reader would demystify this and bring our all your results in the open.**

**I would assume that the large error is something to do with the design of your release experiment and/or in combination with chamber design. There is no real reflection on the representativeness of how you release the CH4 so it simulates emissions at the component level. This was done quite detailed in the soil release studies you refer to where the entire validation of the chamber design hinged on simulating the physical nature of the flux, e.g. diffusion through a porous media. A more detailed discussion of this is needed as it will only increase to the value of your study in terms of application.**

**The large chamber types you use may also be the cause of the large errors, as the collapsible chambers are more susceptible to pressure pumping in the headspace by the wind acting on the walls of the chamber. You do not show results of this nor discuss it, which is really at the core of the applicability and pros/cons of certain chamber types. Here again showing the chamber enclosures and identify the disturbances will help you to discuss this.**

*We agree with the reviewer and have since added additional discussion regarding these experiments with high error. As mentioned above, all of the high error (>40%) experiments occurred in larger volume chambers, with the majority of those not having fans present. We hope that the text added above addresses these concerns.*

**If the above mentioned results and discussion are added to the manuscript I believe the value will be increased a lot.**

---

## Author Comment (AC2)

We would like to thank all reviewers for their valuable insights and comments made to improve this work. We believe we have addressed all comments made by the reviewers as shown below, and the suggested changes have greatly improved this manuscript.

Our response to all comments is structured so that the reviewers comments are shown in "**bold**", our responses are shown in *"italics"*, prior text in the manuscript is shown in "blue", added text is shown in "red", and any deleted text in the manuscript is shown in .

We would once again like to thank the reviewers for their valuable insights and comments regarding this manuscript.

In addition to the changes outlined below for this reviewer, we also performed additional edits that we believe improve on the current version of the manuscript without significantly altering any results. First, we address some previously unmentioned methods we used for the linear regressions for controlled releases where methane concentrations were expected to rapidly reach steady-state. We also modified our presentation of the accuracy of measurements to encapsulate the median of the over- and underestimates to better represent the bias in our measurements. Finally, we incorporated two new studies on prior controlled releases into Figure 2 and the subsequent analysis section. The changes are outlined as follows:

Line 140: "For some experiments, methane concentrations within the chamber were expected to rapidly reach steady-state. Steady-state is reached when methane concentrations no longer increase over time in the chamber and the concentration of methane within the chamber is equal to the concentration of the released gas. The residence time, or time to reach steady-state, is defined by (2): where $\tau$ is the residence time, $V$ is the volume of the chamber, and $Q$ is the volumetric flowrate of gas (i.e., methane and balance gas combined) into the chamber. For any controlled releases where the expected residence time was two minutes or lower, we only used the initial ten data points for the linear regression to avoid the period of exponential decay as methane concentrations approach steady-state (Pihlatie et al., 2013).

(3) $\tau = V/Q$

For each factor being investigated, we grouped the results depending on whether the measurement was an under- or overestimate of the true methane flowrate. We calculated the accuracy of measurements as a range spanning from the median of the over- and underestimated methane flowrates, respectively. We determined the bias of measurements as the average of the raw percentage errors to determine whether tests were biased more towards the under- or overestimation of methane flowrates. "

New Figure 2:

[Figure]

**Reviewer #2**

**This work builds upon previous research on the assessment of the performance of static chamber methods for methane emission quantification. Such controlled release testing is important in interpreting methane emissions measurement data needed to assess the magnitude of emissions and progress toward emission reductions. The focus on component-level measurements is also important, given the methane policy implications. In addition, the detailed assessment of the various factors that could influence measurement accuracy contributes to the novelty of the work presented here. A few comments and suggestions for revisions are included below:**

- **Some additional statistical assessment/visualization could help improve the data interpretation here. A regression analysis/parity plot of the performance of the static chamber method against metered flow rates is standard in these kinds of controlled release experiments, and it is not clear why the authors excluded this in their presentation. These parity plots (and accompanying goodness of fit tests) are easily accessible to**

**the lay person than, say Figure 5, showing the percentage error correlation with metered flow rates. The authors should consider including such statistical analysis and visualization in the revised manuscript.**

*We agree with the reviewer and have since replaced Figure 3 with three new parity plots as suggested. The parity plots also contain histograms of the error distributions. We have also modified Figure 4 to remain a violin plot but instead show the actual measurement errors rather than the absolute measurement errors. Figure 5 has been removed and replaced with a figure showing examples of the raw data from the controlled release experiment for discussion on the large variability in accuracy observed in some of the controlled releases. Examples of the changed figures and revised text are shown below:*

[Figure]

*Parity plot example*

[Figure]

*Revised Figure 4*

[Figure]

*New Figure 8 that replaces old Figure 5*

Line 178-185: "Our analysis of chamber volume with respect to quantification accuracy showed that the accuracy of measurements increased with smaller chamber volumes (Figure 4). The <20 L chambers had the highest accuracy at +12/-12% with an error standard deviation of 12%. The 322 L chamber had a lower accuracy of +15/-17% with a standard deviation of 23%. Our highest errors were measured from the largest 2,265 L chamber with an accuracy of +50/-16% and a standard deviation of 26%. We analyzed all three chamber sizes for bias and found that the <20 L chambers showed a slight tendency for underestimation of flowrates with an average bias of >0%, the 322 L chamber showed a stronger tendency towards the underestimation of flowrates at -18%, and the 2,265 L chamber showed a slight bias towards overestimating flowrates at +7% (Figure 4)."

Line 187-193: "Our comparisons of different chamber shapes showed that the cylindrical chambers were more accurate than the rectangular chambers, showing an accuracy of +5/-14% and a standard deviation of 18% (Figure 5). We found that the rectangular chambers showed a lower accuracy of +17/-14% with a standard deviation of 22%. Similar to the chamber volume, the median percentage error was smaller than the average error for both chamber shapes, which indicates an extreme distribution in percentage errors. We analyzed both chamber shapes for bias and found that the cylindrical chambers were biased towards the underestimation of methane flowrates with an average bias of -13% whereas the rectangular chambers showed a small bias towards the overestimation of methane flowrates with an average bias of +6% (Figure 5)."

Line 195-199: "The most impactful physical factor we observed on chamber measurement accuracy was the presence of fans, where chambers with fans present had a median percentage error of +6/-5% and a standard deviation of 17% (Figure 6), which was higher than chambers without fans which had an accuracy of +17/-17% and a standard deviation of 22%. For both data-sets we observed median values lower than the mean indicating a skewed data-set. We analyzed both data-sets for bias and found that both chambers with and without fans showed slight biases towards the underestimation of methane flowrates at -2% and -4% respectively (Figure 6)."

Line 202-208: "We tested four different mass flowrates for our controlled release tests: 1.02 g/hour, 10.2 g/hour, 102 g/hour, and 511 g/hour (Figure 7). The lowest errors were measured from the 10.2 and 102 g/hour mass flowrates each with accuracies of +8/-11% and +7/-13% respectively. The lowest accuracy of +56/-15% was attributed to the highest mass flowrate of 512 g/hour. We found that the 1.02, 10.2, and 102 g/hour mass flowrates all had negative biases of -11%, -1%, and -6% respectively. The mass flowrate of 512 g/hour had a slight bias of +4% towards the overestimation of mass

flowrates, and also the highest upper accuracy estimate of +46% we observe among the different factors we analyzed."

Line 210-215: "We analyzed six different volumetric flowrates for the range of methane flowrates we tested: 0.238 L/min, 0.476 L/min, 2.38 L/min, 4.76 L/min, 11.9 L/min, and 23.8 L/min (Figure 7). We found that the lowest accuracies were attributed to both the highest and lowest volumetric flowrates with accuracies of +50/-15% and +21/-14% respectively, whereas higher accuracy was observed with the mid-level volumetric flowrates of 11.8, 4.76, 2.38, and 0.476 SLPM with accuracies ranging from +26/-3% to +9/-11%. Similar to the mass flowrates, we also found the highest accuracies were associated with the mid-level volumetric flowrates while the lowest accuracies were observed at the upper and lower volumetric flowrates."

Line 217-222: "We analyzed four different percentages of methane in the leaking gas for the controlled releases (Figure 7). The lowest accuracies were associated with the 5% methane gas with an accuracy of +31/-16%, whereas the highest accuracies were observed with the 10% methane at +15/-8%. The three highest percentages of methane in the leaking gas all had small negative biases ranging from -5% to -3%, whereas the 5% methane leak had a slight positive bias at +1%."

Line 238-250: "At low mass flowrates of methane (i.e., ≤ 100 g/hour), we found that smaller sized chambers were more accurate than larger chambers, with  with accuracies of +12/-8% and +15/-19% respectively. The usage of fans had little impact on the accuracy of smaller sized chambers at these low flowrates, with smaller chambers with fans producing  an accuracy of +16/-8% and smaller chambers without fans having an accuracy of +7/-13% . In contrast,  the usage of fans was important for the accuracy of larger chambers at  lower mass flowrates. Larger chambers with fans  had an accuracy of +4/-30% and larger chambers without fans had an accuracy of +48/-19%. In terms of chamber shape,  at low flowrates smaller cylindrical chambers had an accuracy of +1/-11% compared to small rectangular chambers which produced an accuracy of +15/-3% . For larger chambers at low mass flowrates, we  observed a contrasting result with large rectangular chambers producing an accuracy of +6/-16% and large cylindrical chambers producing a median percentage error of +24/-48%.

Line 251-258: "We observed similar results for optimizing chamber configurations for high methane mass flowrates (i.e., ≥100 g/hour). We found  that the usage of fans was critical for measurement accuracy for larger sized chambers at these higher mass flowrates of methane. Larger chambers with fans had an accuracy of +4/-4%  compared to larger chambers without fans which had had an accuracy of +66/-35% . For chamber shapes,  cylindrical chambers were more accurate than rectangular chambers with an accuracy of +6/-14% compared to +26/-15% from rectangular chambers . At  higher mass flowrates of methane, we found that large cylindrical chambers were highly accurate at +2/-3% of the true methane flowrate . "

- **In the discussion section (e.g., Page 13, lines 289-296), some more direct comparison with previous studies could be useful. Is a median percentage error of 14% consistent with similar previous studies? Similarly, how does the quantification performance of the static chamber method as assessed here compares with other indirect quantification methods?**

*This is an excellent suggestion, and we have now added some additional text in the discussion that comments on previous controlled release experiments and the expected quantification errors. We chose one paper for mobile surveys and one for aircraft based surveys, which gives a rough idea of where the accuracy of indirect methods lies.*

Line 290: "Our results show that the static chamber methodology can quantify methane emissions ranging from 1.02 g/hour to 512 g/hour with a median percentage error of ±14%.  In

comparison to indirect methods,  Johnson et al., 2023 state that their aircraft-based method has a multi-pass uncertainty range of -46/+54%, which roughly corresponds to an absolute error of ±50%. In von Fischer et al., 2017, they state an uncertainty range of -24/+32% after five mobile survey passes, which roughly corresponds to an absolute error of ±28%. With regards to other controlled release tests on static chambers, we do find that our median uncertainty of ±14% falls within the 10-20% range reported by Lebel et al. 2020 and Pihlahtie et al. 2013."

- **On Page 13, lines 297 to 307, it is not clear, and was not tested here, whether static chamber methods can accurately quantify emission rates beyond e.g., greater than 1,000 g/h. Is there a threshold beyond which this method does not work? If so, there needs to be a discussion of that limitation here, otherwise the reader is left with the impression that this method can be used to quantify all kinds of emission rates > 100 g/h.**

*Theoretically there is no upper limit, although practically the upper limit would be somewhere in the range of 250 kg/hour based on the largest chamber ever used in literature (~32,000 L by Lebel et al., 2020). Given a methane flowrate of 250 kg/hour, it would take ~5 minutes for a 32,000 Litre chamber to reach steady-state. We have since added additional text to the manuscript to reflect this limitation.*

Line 300-303: "Methane concentrations within a smaller chamber can also rapidly reach explosive levels which can pose safety concerns if the environment is not intrinsically safe (Riddick et al., 2022), but these risks can be minimized at little cost to accuracy if fans are omitted. Furthermore, intrinsically safe methods of chamber mixing such as external pumps could be used to mix air within chambers, regardless of the size of chamber. Theoretically, there is no upper methane flowrate limitation of the static chamber method, and utilizing large chambers such as the ~32,000 L chamber used in (Lebel et al., 2020) could theoretically quantify methane flowrates in the 100-200 kg/hour range. However, there are practical limitations to directly measuring components emitting methane at these high levels, the most notable being safety concerns and access issues (e.g., measuring flare stacks and liquid storage tank unloadings). Another factor to consider is the time to reach steady-state. Enclosing a high methane emitting source within a smaller chamber causes methane concentrations within the chamber to rapidly reach steady-state, essentially creating a dynamic chamber, which we do not test in this work (Pedersen et al., 2010, Levy et al., 2011)."

- **The visualization of the percent error (Equation 2, Figures 3-5) is potentially misleading. Because these are presented on an absolute basis using equation 2, Figures 3-5 could be interpreted that all quantified emission**

**rates are greater than metered rates. One could assume that there were quantified emission rates that were less than metered rates, which would lead to a negative percent error, in some cases. The authors should consider revising Figures 3-5 to include those values that were quantified both above (positive percent errors) and below (negative percent errors) the actual metered emission rates.**

*We have since changed these figures as mentioned in an earlier comment to this reviewer. We hope the new changes address these concerns.*

- **Line 19-20, please include a reference for the Global Methane Pledge.**

*Done, reference added.*

- **In the introduction, suggest including specific examples of "components" that can be quantified using the static chamber method (e.g., wellheads at oil and gas well sites). Also, can static chamber methods quantify total facility level emissions, which could be thought of as an aggregation of emissions from individual methane emitting "components" at the facility?**

*Agreed, we have added examples of different components in the Introduction. We have also added a sentence that the static chamber method would not be able to quantify facility-level emissions through a single measurement, but could if enough measurements from sites/components are made. Although, there is always a potential to "miss" an emitting component/site using direct measurement techniques, which we have already highlighted in the text.*

Lines 45-48: "Methane sources can be classified as component, site, facility, regional, and global level sources in order of increasing spatial scales (NACEM, 2018). As an example, a valve on an abandoned oil and gas well would constitute a component level source whereas all abandoned oil and gas wells in the Appalachian basin would comprise a regional methane source. The advantages of methane inventories created from component level measurements are high resolution and easy comparisons to regional inventories ,which are predominantly made using component level data (U.S. GHGI, ECCC GHGI), where specific discrepancies can be identified (Rutherford et al., 2021). Indirect measurements can be used to measure  methane emissions at site/facility/regional levels, but have higher limits of detection when compared to direct methods and additional challenges related to source attribution at the component level. On the other hand, direct measurement methods are labour intensive and  can omit methane sources when scaling up measurements to

facility/regional/global levels , but can quantify and attribute methane emissions at the component level."

---

## Author Comment (AC3)

We would like to thank all reviewers for their valuable insights and comments made to improve this work. We believe we have addressed all comments made by the reviewers as shown below, and the suggested changes have greatly improved this manuscript.

Our response to all comments is structured so that the reviewers comments are shown in "**bold**", our responses are shown in *"italics"*, prior text in the manuscript is shown in "blue", added text is shown in "red", and any deleted text in the manuscript is shown in "blue strikethrough".

We would once again like to thank the reviewers for their valuable insights and comments regarding this manuscript.

In addition to the changes outlined below for this reviewer, we also performed additional edits that we believe improve on the current version of the manuscript without significantly altering any results. First, we address some previously unmentioned methods we used for the linear regressions for controlled releases where methane concentrations were expected to rapidly reach steady-state. We also modified our presentation of the accuracy of measurements to encapsulate the median of the over- and underestimates to better represent the bias in our measurements. Finally, we incorporated two new studies on prior controlled releases into Figure 2 and the subsequent analysis section. The changes are outlined as follows:

Line 140: "For some experiments, methane concentrations within the chamber were expected to rapidly reach steady-state. Steady-state is reached when methane concentrations no longer increase over time in the chamber and the concentration of methane within the chamber is equal to the concentration of the released gas. The residence time, or time to reach steady-state, is defined by (2): where $\tau$ is the residence time, $V$ is the volume of the chamber, and $Q$ is the volumetric flowrate of gas (i.e., methane and balance gas combined) into the chamber. For any controlled releases where the expected residence time was two minutes or lower, we only used the initial ten data points for the linear regression to avoid the period of exponential decay as methane concentrations approach steady-state (Pihlatie et al., 2013).

(3) $\tau = V/Q$

For each factor being investigated, we grouped the results depending on whether the measurement was an under- or overestimate of the true methane flowrate. We calculated the accuracy of measurements as a range spanning from the median of the over- and underestimated methane flowrates, respectively. We determined the bias of measurements as the average of the raw percentage errors to determine whether tests were biased more towards the under- or overestimation of methane flowrates. "

New Figure 2:

[Figure]

**Reviewer #3**

**This was a very interesting read and the author shows the value of improving methane emission quantification at the component level. The paper compares previously used emission factors and evaluates previous controlled direct and indirect controlled release tests.  The study also examines direct measurements techniques from static chambers for estimating methane emission rates. These types of techniques have solely been tested and examined in detail which provides useful information for scientists studying methane emissions looking for the right techniques.**

**In general, the paper is a nice contribution to this research area.  The writing and structure is clear and well organized, but "we found" can be used less.  The paper could use more visuals and photographs of the tested experiments and measurements.  Overall, I approve of this paper with the minor revisions:**

*We thank the reviewer for the valuable insights and comments regarding this work. With regards to the excessive use of "we found", we have modified the text for better flow for the*

*reader which we hope improves this work. The outlined changes in that regard are outlined below:*

Line 178-185: "Our analysis of chamber volume with respect to quantification accuracy showed that the accuracy of measurements increased with smaller chamber volumes (Figure 4). The <20 L chambers had the highest accuracy at +12/-12% with an error standard deviation of 12%. The 322 L chamber had a lower accuracy of +15/-17% with a standard deviation of 23%. Our highest errors were measured from the largest 2,265 L chamber with an accuracy of +50/-16% and a standard deviation of 26%. We analyzed all three chamber sizes for bias and found that the <20 L chambers showed a slight tendency for underestimation of flowrates with an average bias of >0%, the 322 L chamber showed a stronger tendency towards the underestimation of flowrates at -18%, and the 2,265 L chamber showed a slight bias towards overestimating flowrates at +7% (Figure 4)."

Line 187-193: "Our comparisons of different chamber shapes showed that the cylindrical chambers were more accurate than the rectangular chambers, showing an accuracy of +5/-14% and a standard deviation of 18% (Figure 5). We found that the rectangular chambers showed a lower accuracy of +17/-14% with a standard deviation of 22%. Similar to the chamber volume, the median percentage error was smaller than the average error for both chamber shapes, which indicates an extreme distribution in percentage errors. We analyzed both chamber shapes for bias and found that the cylindrical chambers were biased towards the underestimation of methane flowrates with an average bias of -13% whereas the rectangular chambers showed a small bias towards the overestimation of methane flowrates with an average bias of +6% (Figure 5)."

Line 195-199: "The most impactful physical factor we observed on chamber measurement accuracy was the presence of fans, where chambers with fans present had a median percentage error of +6/-5% and a standard deviation of 17% (Figure 6), which was higher than chambers without fans which had an accuracy of +17/-17% and a standard deviation of 22%. For both data-sets we observed median values lower than the mean indicating a skewed data-set. We analyzed both data-sets for bias and found that both chambers with and without fans showed slight biases towards the underestimation of methane flowrates at -2% and -4% respectively (Figure 6)."

Line 202-208: "We tested four different mass flowrates for our controlled release tests: 1.02 g/hour, 10.2 g/hour, 102 g/hour, and 511 g/hour (Figure 7). The lowest errors were measured from the 10.2 and 102 g/hour mass flowrates each with accuracies of +8/-11% and +7/-13% respectively. The lowest accuracy of +56/-15% was attributed to the highest mass flowrate of 512 g/hour. We found that the 1.02, 10.2, and 102 g/hour

mass flowrates all had negative biases of -11%, -1%, and -6% respectively. The mass flowrate of 512 g/hour had a slight bias of +4% towards the overestimation of mass flowrates, and also the highest upper accuracy estimate of +46% we observe among the different factors we analyzed."

Line 210-215: "We analyzed six different volumetric flowrates for the range of methane flowrates we tested: 0.238 L/min, 0.476 L/min, 2.38 L/min, 4.76 L/min, 11.9 L/min, and 23.8 L/min (Figure 7). We found that the lowest accuracies were attributed to both the highest and lowest volumetric flowrates with accuracies of +50/-15% and +21/-14% respectively, whereas higher accuracy was observed with the mid-level volumetric flowrates of 11.8, 4.76, 2.38, and 0.476 SLPM with accuracies ranging from +26/-3% to +9/-11%. Similar to the mass flowrates, we also found the highest accuracies were associated with the mid-level volumetric flowrates while the lowest accuracies were observed at the upper and lower volumetric flowrates."

Line 217-222: "We analyzed four different percentages of methane in the leaking gas for the controlled releases (Figure 7). The lowest accuracies were associated with the 5% methane gas with an accuracy of +31/-16%, whereas the highest accuracies were observed with the 10% methane at +15/-8%. The three highest percentages of methane in the leaking gas all had small negative biases ranging from -5% to -3%, whereas the 5% methane leak had a slight positive bias at +1%."

Line 225-226: "We  analyzed how  chamber configurations (i.e., chamber volume, usage of fans, chamber shapes) can be optimized…"

Line 229: " We also  saw that smaller chambers performed similarly if fans…"

Line 231: "Smaller chambers performed slightly better…"

Line 233: "For larger sized chambers (i.e., ≥20L),  the usage of fans was critical…"

Line 238-250: "At low mass flowrates of methane (i.e., ≤ 100 g/hour), we found that smaller sized chambers were more accurate than larger chambers, with  with accuracies of +12/-8% and +15/-19% respectively. The usage of fans had little impact on the accuracy of smaller sized chambers at these low flowrates, with smaller chambers with fans producing  an accuracy of +16/-8% and smaller chambers without fans having an accuracy of +7/-13% . In contrast,  the usage of fans was important for the accuracy of larger chambers at  lower mass flowrates. Larger chambers with fans

 had an accuracy of +4/-30% and larger chambers without fans had an accuracy of +48/-19%. In terms of chamber shape,  at low flowrates smaller cylindrical chambers had an accuracy of +1/-11% compared to small rectangular chambers which produced an accuracy of +15/-3%  For larger chambers at low mass flowrates, we  observed a contrasting result with large rectangular chambers producing an accuracy of +6/-16% and large cylindrical chambers producing a median percentage error of +24/-48%.

Line 251-258: "We observed similar results for optimizing chamber configurations for high methane mass flowrates (i.e., ≥100 g/hour). We found  that the usage of fans was critical for measurement accuracy for larger sized chambers at these higher mass flowrates of methane. Larger chambers with fans had an accuracy of +4/-4%  compared to larger chambers without fans which had had an accuracy of +66/-35% . For chamber shapes,  cylindrical chambers were more accurate than rectangular chambers with an accuracy of +6/-14% compared to +26/-15% from rectangular chambers . At  higher mass flowrates of methane, we found that large cylindrical chambers  were highly accurate at +2/-3% of the true methane flowrate . "

**Line 26:  add a more detailed description of what component level means or provide distinct examples.**

*We have added new text in the introduction that better defines the various spatial scales and provided examples.*

Lines 45-48: "Methane sources can be classified as component, site, facility, regional, and global level sources in order of increasing spatial scales (NACEM, 2018). As an example, a valve on an abandoned oil and gas well would constitute a component level source whereas all abandoned oil and gas wells in the Appalachian basin would comprise a regional methane source. The advantages of methane inventories created from component level measurements are high resolution and easy comparisons to regional inventories ,which are predominantly made using component level data (U.S. GHGI, ECCC GHGI), where specific discrepancies can be identified (Rutherford et al.,

2021). Indirect measurements can be used to measure  methane emissions at site/facility/regional levels, but have higher limits of detection when compared to direct methods and additional challenges related to source attribution at the component level. On the other hand, direct measurement methods are labour intensive and  can omit methane sources when scaling up measurements to facility/regional/global levels , but can quantify and attribute methane emissions at the component level."

**Line 41: list the recent paper on satellite estimations by de Foy et al., 2023 (DOI 10.1088/1748-9326/acc118).**

*Excellent reference, thank you for the suggestion. We have added added to introduction.*

**Lines 100-135: A visual would be useful for the reader here. I would suggest a general diagram or image of the static chambers with the main components and also a schematic or field photo of how the controlled release was set-up. The author could also include a material list and description of the construction in the SI with what each component/material was used for and why.**

*Agreed, we have since added a new figure 1 that shows the general set-up as well as photos of all chambers in field and test settings.*

[Figure]

**Discussion: It would be beneficial to include more comparison to the other static**

**chamber studies mentioned in the introduction and describe how the results from this study could explain outcomes in the other studies.**

*Unfortunately the studies aren't exactly comparable since the only two static chamber studies with documented uncertainties (Pilahtie et al. and Christiansen et al.) focused on relativelysmall emission rates compared to the ranges we tested, and they also focused specifically on soil gas emissions whereas our focus is on component sources. Another recent study (Riddick et al. 2022) withdrew their static chamber results from their submission, so we could not compare to that study either. However, we can compare with Lebel et al. who mention in their SI that the static chamber method underestimated methane flowrates by 10-20% on average based on their tests (flowrates not reported unfortunately). We have added some text in the Discussion to reflect this.*

Line 290: "Our results show that the static chamber methodology can quantify methane emissions ranging from 1.02 g/hour to 512 g/hour with a median percentage error of ±14%. In comparison to indirect methods, Johnson et al., 2023 state that their aircraft-based method has a multi-pass uncertainty range of -46/+54%, which roughly corresponds to an absolute error of ±50%. In von Fischer et al., 2017, they state an uncertainty range of -24/+32% after five mobile survey passes, which roughly corresponds to an absolute error of ±28%. With regards to other controlled release tests on static chambers, we do find that our median uncertainty of ±14% falls within the 10-20% range reported by Lebel et al. 2020 and below the 33% estimated from Pihlahtie et al. 2013 for the linear calculator method."